# Changes in the Oxidative Status of Dual-Purpose Hens Rearing in the Free-Range System during Cold, Thermoneutral and Hot Period

**DOI:** 10.3390/ani12192650

**Published:** 2022-10-02

**Authors:** Vasko Gerzilov, Albena Alexandrova, Petar Petrov, Veselina Boncheva, Neli Keranova, Madlena Andreeva, Almira Georgieva, Elina Tzvetanova

**Affiliations:** 1Department of Animal Science, Agricultural University, 12, Mendeleev Str., 4000 Plovdiv, Bulgaria; 2Laboratory of Free Radical Processes, Institute of Neurobiology, Bulgarian Academy of Sciences, 23, Acad. G. Bonchev Str., 1113 Sofia, Bulgaria; 3Department of Physiology & Biochemistry, National Sports Academy, 21, Acad. S. Mladenov Str., 1700 Sofia, Bulgaria; 4Department of Mathematics & Informatics, Agricultural University, 12, Mendeleev Str., 4000 Plovdiv, Bulgaria

**Keywords:** antioxidant enzymes, chicken genotype, cold and hot periods, oxidative status, welfare

## Abstract

**Simple Summary:**

Nowadays, most consumers prefer table eggs from free-range laying hens. At the same time, relatively few studies exist on the possible effects of free-range rearing on the oxidative status of fowls and their egg production. This trial studied the influence of environmental conditions (cold, thermoneutral and hot period) on the oxidative status of different genotypes of free-range laying hens and the effect of oxidative stress (OS) on their egg production. The factor “temperature period”, compared to “year” and “genotype”, had the most significant influence on all biochemical parameters determining OS. The chicken genotypes showed differences in their susceptibility to OS, and this had an effect on egg production. The OS is genotypically specific and can play a significant role in determining welfare and egg production in free-range systems.

**Abstract:**

This study aimed to assess the changes in the oxidative status of six genotypes of free-range laying hens during cold, thermoneutral, and hot periods by measuring the levels of lipid peroxidation (LPO), total glutathione (tGSH), and the activity of antioxidant enzymes catalase (CAT), superoxide dismutase (SOD), and glutathione peroxidase (GPx) in erythrocyte suspension, in relation with their egg production. Two identical experiments were conducted in two consecutive years. Thermal stress adversely affected the oxidative status of hens. The induced OS is expressed by an increase in LPO and the activities of antioxidant enzymes SOD and GPx during cold and hot periods and a decrease in CAT and tGSH during the cold period in both years. The factor “temperature period”, compared to “year” and “genotype”, had the most significant influence on all biochemical parameters (*p* < 0.001). Significant phenotypic correlations (*p* < 0.05) were detected among studied biochemical parameters, except between SOD and tGSH. The chicken genotypes showed differences in their susceptibility to OS and this had an effect on egg production—from 37.87% to 74.93%. The OS is genotypically specific and can play a significant role in determining welfare and egg production in free-range systems.

## 1. Introduction

As a result of consumers’ changing perception of animal production systems, there has been an increased interest in free-range poultry production. It is believed that free-range birds have specific advantages, such as direct contact with sun rays and fresh air, opportunity to perform their natural behavioural instincts (rooting, dust bathing), pecking grains, or pebbles [1]. On the other hand, a higher risk of stress exists in free-range rearing systems, associated with factors as changing environmental conditions, parasitic, bacterial and viral infections [2,3], contamination, aggression from dominant birds, pecking and risk of cannibalism [4], predators [5], etc., and, in particular, ambient temperature is being widely recognized as a main stress factor [6,7,8]. It seems highly realistic that free-range farming may also face problems associated with local manifestations of climate change, especially the extension of the warm period not only in geographic areas affected by high environmental temperatures but also in Europe [9,10].

Stress is a response of the body to adverse stimuli, which is nonspecific, and any agent that induces stress is defined as a stressor [11]. Therefore, any type of stress exhibits a biological response or reaction to stimuli that affect normal physiological homeostasis. OS can be triggered by a number of factors as a reaction to their impact. OS is a condition characterized by an imbalance between the pro-oxidant processes, induced by the reactive oxygen species (ROS), and the antioxidant defence in organisms [12]. ROS are highly reactive and can threaten the cellular constituents, leading ultimately to cell death. Because ROS are normally generated in the organism, a defence system (including enzyme and non-enzyme antioxidants) was developed evolutionarily in order to neutralize their harmful effects. However, upon different environmental stressors (as extreme temperatures), excess production of ROS is provoked that exceeds the defence mechanisms capacity and OS can occur [6]. Mild stress serves as a signal to activate signalling pathways important to overcome the effects of negative factors and to ensure normal physiological functions. Severe stress, however, leads to cell structure damage and cell death.

In controlled trials, the chronic or acute exposure of birds to high or low temperatures increased lipid peroxidation (LPO) levels, as a marker of OS, and changes in the activity of antioxidant enzymes were established [7,13,14,15]. In turn, increased LPO can have a significant negative impact on the well-being of the birds and, accordingly, on the quantity and quality of the expected products [16].

The conventional cage systems for laying hens have been banned, at least in the European Union [17] and more farms turn towards free-range rearing systems. At the same time, relatively few studies exist on the possible effects of free-range rearing on the oxidative status of birds and their performance. Although the interest in free-range farming is rising, there is still relatively little research concerning the adaptability of the used chicken genotypes to variations in environmental factors during the free-range rearing and impacts on the hen’s productivity. As far as different rearing conditions are known to cause different stress, an interesting question arises, concerning whether different genotypes of free-range layers are differently susceptible to OS and how this can affect egg production. Some studies suggest that different chicken’s genotypes may be differentially sensitive to OS, and this may affect their liveability and performance [18,19].

Given the ability of OS to direct the cellular response in different directions and the genetic predisposition of the functional qualities of organisms, we hypothesize that shifts in the pro/antioxidant balance in chickens may depend more strongly, than previously known, on the genotype of the birds, which can be manifested on the physiological level by significant changes in their performance in response to rearing conditions. Thus, the aim of the present study was to assess the changes in the oxidative status of different chicken’s genotypes in response to the conditions of the environment in a free-range rearing system and the possible effects on egg production performance. The changes in oxidation status were reported by measuring the levels of the following biochemical parameters: LPO, tGSH, CAT, SOD, and GPx in erythrocyte suspension.

## 2. Materials and Methods

### 2.1. Birds and Housing

In the present study, six well-known genotypes of dual-purpose chickens were used, which were reared under a free-range system. Tetra H and Tetra Super Harco are hybrid combinations produced by Bábolna Tetra Kft. (Hungary) for industrial laying purposes. White Plymouth Rock (line G) and Barred Plymouth Rock (line E) are lines from the National Gene Pool of Bulgaria, which, together with the Australorp (Australia), are well known and widespread breeds that are very popular in small farms due to their high egg yield. The Bielefelder is a relatively new breed originating from Germany.

Chickens were divided into six groups according to their genotype, each including 27 females and 3 males, and were reared from 19 to 52 weeks of age. The experiment was carried out in two consecutive years, i.e., the experiment from the first year was repeated in the next year. All groups were reared under the same conditions during the experiment: in pen (size 3.50/2.50/2.75 m) with pop-holes and free access to a fenced yard (size 9.20/24 m) in the Poultry division of the Agricultural University—Plovdiv, Bulgaria (Figure 1). All routine and occasional management practices (vaccination and medication) were strictly adhered to.

Compound feed (mash form) and water were offered for consumption *ad libitum*. The diets were with equal feed and nutrient composition for all groups and were consistent with the laying phase—Table 1.

Egg production was monitored from the age at first egg to 52 weeks of age (in % weekly). The eggs were collected daily and the total number recorded, with respect to the group. Hen—week egg production (%) was determined as:

Total number of eggs produced by a flock per week × 100/total number of hens housed × 7.

Meteorology data were taken from the Agro-meteorology station at the National Institute of Meteorology and Hydrology—Branch Plovdiv, located immediately next to the Poultry division of the Agricultural University of Plovdiv (latitude 42°14′ N, longitude 24°79′ E, and altitude of 164 m above sea level).

### 2.2. Ethics Statement

All experimental procedures were performed by the rules of the Animal Ethics Committee at the Agricultural University—Plovdiv, as well as Ordinance № 20 of 1.11.2012 on the minimum requirements for protection and welfare of experimental animals and site requirements for use, cultivation, and/or their delivery [20].

### 2.3. Oxidative Status Determination—Blood Collection and Biochemical Analysis

The changes in the oxidative status of the hens from the studied six genotypes were evaluated in erythrocyte suspensions. Blood samples (*n* = 6) from each group of laying hens were taken during three thermal periods: cold, neutral, and hot in the 1st and 2nd experimental years when the birds were at the 36th, 44th, and 52nd weeks. The ambient temperatures during blood sampling are presented in Table 2.

The blood of the birds was taken from the *v. subcutanea ulnaris* in vacutainers (BD–Plymouth, PL6 7BP, UK), as the duration of the manipulation did not exceed 2 min.

The blood was centrifuged at 600× *g* for 10 min at 4 °C. The plasma was removed from the red blood cells and the latter were washed twice with physiological saline under the same conditions. The obtained erythrocyte suspension was frozen and stored at −80 °C until analysis.

Erythrocytes as a 5% suspension (diluted according to Hb concentration in 0.15 M NaCl—10 mM sodium phosphate buffer, pH 7.2) was used for measurement of the level of lipid peroxidation (LPO), total glutathione (tGSH), and the activity of antioxidant enzymes: catalase (CAT), superoxide dismutase (SOD), and glutathione peroxidase (GPx).

Hemoglobin (Hb) amount in erythrocyte suspension was determined spectrophotometrically, using Drabkin’s reagent. A total of 10 μL 5% erythrocyte suspension was added into Drabkin reagent (total volume 1 mL) and after 20 min incubation at room temperature the absorbance at 540 nm was read against Drabkin reagent. The total haemoglobin concentration (g/L) was determined using a calibration curve.

The levels of LPO, tGSH, and the activities of the antioxidant enzymes SOD, CAT, and GPx were determined spectrophotometrically using commercially available kits purchased from Sigma-Aldrich Co., LLC (St. Louis, MO, USA): Lipid peroxidation (MDA) Assay Kit MAK085, Glutathione Assay Kit CS0260, SOD Assay Kit-WST 19160, Catalase Assay Kit CAT100, Glutathione Peroxidase Cellular Activity Assay CGP1. The manufacturer’s instructions were strictly followed for all procedures.

### 2.4. Statistical Analysis

Data analysis on the OS markers was performed using IBM Statistics SPSS 24 [21,22]. A three-way ANOVA analysis was selected to investigate the effects of different factors (year, period, and genotype) and their interactions on the OS markers. When the differences between the variants were significant, the Tukey HSD test for post hoc analysis of means was performed (*p* < 0.05). Phenotypic correlations between OS markers were analysed using Pearson’s correlation coefficient.

The statistical model employed was:W_ijk_ = μ + Y_i_ + P_j_ + G_k_ + Y_i ×_ P_j_ + Y_i ×_ G_k_ + P_j ×_ G_k_ + Y_i ×_ P_j ×_ G_k_ + ε_ijk_(1)
where:

W_ijk_—individual observation;

μ—population mean;

Y_i_—effect of ith year (i = 2; first and second year);

P_j_—effect of jth period (j = 3; cold, thermoneutral and hot periods);

G_k_—effect of kth genotype (k = 6; Tetra H, Tetra Super Harco, White Plymouth Rock, Barred Plymouth Rock, Bielefelder, Australorp);

Y_i ×_ P_j_—effect of the interaction of ith year and jth period;

Y_i ×_ G_k_—effect of the interaction of ith year and kth genotype;

P_j_ × G_k_—effect of the interaction of jth period and kth genotype;

Y_i ×_ P_j ×_ G_k_—effect of the interaction of ith year, jth period and kth genotype;

ε_ijk_—random residual error normally and independently distributed with zero mean and common variance.

## 3. Results

The results of the experimentally assayed OS markers in the hens from the studied genotypes reared in the free-range system are presented in Table 3. The data (m ± SEM) showed high variability in the level of the different markers.

It was found that the lowest values of LPO were present in the hens during the thermoneutral period for both years. The lowest LPO value (0.16 nmoles MDA/mg Hb) was observed in the Australorp breed during the thermoneutral period in the 1st year. It should be noted that in this genotype the LPO was low, regardless of the period. Both the cold and hot periods appeared to be unfavourable with elevated LPO in all genotypes. The highest LPO values were recorded for the Tetra Super Harco during the cold period of the 2nd year (1.98 nmoles MDA/mg Hb) and also for Barred Plymouth Rock (1.91 nmoles MDA/mg Hb) and White Plymouth Rock (1.91 nmoles MDA/mg Hb) during the hot period of the 1st year. Specific variations in response to the rearing conditions were also manifested by the tGSH concentrations in the erythrocyte suspension. Significantly lower tGSH levels were observed in the hens during the cold period, compared to the thermoneutral and hot periods (*p* < 0.05). The highest value was recorded during the thermoneutral period of the 2nd year for Withe Plymouth Rock (1282 ng/mg Hb) and the lowest was found in the Australorp breed during the cold period of the 1st year (490 ng/mg Hb).

Variations in the antioxidant enzyme system in relation to the rearing conditions were also well demonstrated. CAT activity for all genotypes tested was lower during the cold period and higher during the hot period, in comparison to the thermoneutral period of both year replicates. The highest value was recorded in Tetra Super Harco during the hot period of the 1st year (1.54 U/mg Hb) and the lowest in Barred Plymouth Rock during the cold period of the same year (0.23 U/mg Hb). The estimated SOD activities were higher both in the cold and the hot periods, indicating that low and high temperatures lead to activation of the enzyme with a more expressed increase during the hot period. The maximum SOD value (5.51 U/mg Hb) was observed for Tetra H during the hot period of the 1st year, followed by Tetra Super Harco for the same period (4.87 U/mg Hb). The analysis of the changes in GPx activity showed the same tendency as for SOD. In comparison to the thermoneutral period, an increase in the enzyme activity was observed during the cold and hot periods of both years in all chicken genotypes, except for Bielefelder and Australorp breeds during the hot period. Tetra Super Harco had the lowest enzyme activity with values of 16.75 U/mg Hb and 13.50 U/mg Hb during the thermoneutral period in the 1st and 2nd year, correspondingly.

A correlation analysis was performed to determine the degree of correlation between the individual indicators. The calculated phenotypic correlation coefficients are presented in Table 4. The only unproven relationship is between SOD and tGSH. Between all other indicators, significant phenotypic correlations (r) are reported. The increase in LPO has a slight and negative impact on tGSH (r = −0.262 **) and low-to-moderate positive impact on CAT (r = 0.228 **), SOD (r = 0.402 **) and GPx (r = 0.200 **). The phenotypic correlation between tGSH and CAT is weakly positive (r = 0.224 **) and negative between tGSH and GPx (r = −0.169 *). Increases in CAT values have a moderately positive effect on the change in SOD (r = 0.477 **) and weak effect on that of GPx (r = 0.155 *). There is also a low phenotypic correlation between SOD and GPx (r = 0.164 *).

In an attempt to assess the overall effects of the changing rearing condition factors (year, period, and genotype) on the biochemical indicators of OS (LPO, tGSH, CAT, SOD, GPx) a factorial ANOVA was carried out (Table 5). All studied factors proved to have statistically significant main effects on the OS indicators (*p* < 0.05), except for the factor “year” on LPO and “genotype” on tGSH (*p* > 0.05). In addition, all interactions of the studied factors proved to have significant effects.

The statistically significant effects of the “year” (both main effects and interactions with the other factors) on the variation of OS markers confirmed that the factors of the rearing environment (i.e., primarily the seasonal temperature variation) were different in the two years replicates, although the experimental design was the same.

The studied chicken genotypes are significantly different in terms of egg productivity (Figure 2 and Figure 3). The Bielefelder breed seemed to be the most susceptible to high temperature, as in the first year its production in June was less than 30% (twice less than of the hens from the other genotypes). In general, in both year replicates, the hens belonging to the Tetra Super Harco had the highest egg production (63.28% during the 1st year and 74.93% during the 2nd year). In the 1st year, the second highest egg production belonged to the Australorp breed (58.89%) and to the two chicken genotypes from the National Genetic Pool of Bulgaria: Barred Plymouth Rock (line E)—57.69%, and White Plymouth Rock (line G)—54.18%. In the 2nd year, the third highest egg production belonged to the Barred Plymouth Rock breed (64.96%) after White Plymouth Rock (67.18%) and Tetra Super Harco (74.93). The Bielefelder breed was the lowest egg production in both years—37.87% and 51.93%, respectively.

## 4. Discussion

The main aim of the present research was to investigate the effects of seasonal variations in the rearing conditions in a free-range system on the pro/antioxidant balance in six genotypes of chickens, as defined by levels of the markers LPO, tGSH, CAT, SOD, and GPx. The effect of the oxidative status on the performance of the studied chicken’s genotypes was also assessed. There are data that show the activity of some antioxidant enzymes, and the oxidative stability of chickens are genetically determined and correlate with some production characteristics [23,24].

Dual-purpose chickens are preferred for small-scale poultry production farming because they are both excellent layers and good meat producers. Although free-range rearing has a number of acknowledged advantages [1], there are also a number of occurring unfavourable circumstances or conditions [2,3,5,6,7] that can affect the pro-antioxidant balance in chickens [13,25,26,27]. Accordingly, the wellness of birds is related to their performance and in recent years a number of studies reported the effects of poultry rearing systems on egg production and quality, as well as on their OS [18,28].

Results of the present study clearly showed that the blood pro/antioxidant balance of the laying chicken’ genotypes was affected by changes in the environmental factors of the rearing period, in particular, seasonal temperature changes. Our results demonstrated variability in the values of different markers in response to the temperature period and the genotype. The comparative analysis showed that in the samples taken during both the cold and hot periods, a three to six-fold increase in erythrocyte LPO levels was observed in all genotypes, in comparison to the thermoneutral period. The effects of the seasonal changes of the rearing temperatures in the two year replicates and the genotype on the OS markers were statistically significant both as main effects and also as interactive effects (Table 4). These results are consistent with the findings of other authors reporting similar effects in chickens’ organs and tissues in controlled trials and reared in free environmental conditions with low [13,15] and high temperatures [29,30]. Specifically, in our study during the cold rearing conditions, the highest level of the pro-oxidant marker LPO was induced in Tetra Super Harco hens, in comparison to the other groups, whereas the Bielefelder breed had the lowest LPO values measured. Thus, this genotype appears to be cold resistant. The least affected by the heat impact in regard to LPO induction were the hens belonging to the Australorp breed. The observed increase in LPO during the cold period compared to the thermoneutral one was accompanied by a decrease in the tGSH level in the erythrocyte suspension of all tested genotypes. Reduced glutathione is an essential part of the body’s antioxidant defence system being major cellular non-enzymatic antioxidant and a limiting co-substrate of the antioxidant enzyme GPx. Only in its presence GPx catalyses the reduction of H_2_O_2_ or organic peroxides [31]. Plasma and erythrocyte levels of tGSH is one of the main indicators of oxidative-reductive processes in the body. Its depletion in the birds is associated with the development of OS [32]. With regard to the antioxidant enzymes assessed in this study, an increase in the erythrocyte SOD and GPx activity during the cold and hot periods, compared to the thermoneutral period was observed. A similar activation of antioxidant enzymes has been described for chickens by many authors [15,33,34]. The activation of antioxidant enzymes is an adaptive response, aiming to protect cells from excess ROS generation, induced by the unfavourable environmental conditions [35]. The CAT activities showed a decrease during the cold periods and increase during the hot periods in both years. These variations in the antioxidant enzyme responses are not surprising, since the exposure to acute or prolonged unfavourable conditions can lead to the enzymes inhibition. This is explained by the fact that antioxidant enzymes usually exhibit a “bell-shaped” response to increasing severity or exposure time of the pro-oxidant effects [36]. Thus, it can be assumed that the observed low values of LPO in the Australorp breed are really caused by the observed relatively high activities of the antioxidant enzymes—SOD, CAT, and GPx.

Based on the comprehension that antioxidants in the body provide conditions for adaptive homeostasis [37] and the obtained results and reasoning made above, it can be assumed that hens from the Australorp breed are better adapted to the cold period, judging by the high activities of CAT and GPx recorded in this period in both studied years, while the hens from Tetra H are better adapted to the warm period with the highest activities of SOD, CAT, and GPx reported then in both studied years.

A significant research question concerned the presence of relationships between the OS levels and the egg production of different hen’s genotypes, reared under free-range rearing systems. Previous studies have shown the presence of a reverse relation between the level of LPO, as a pro-oxidative marker, and egg production [38]. However, in our study, significant direct correlation between LPO and egg productivity was not found. Taking into account the complex interactions among pro/antioxidant processes in determining OS levels, we applied factorial analysis of variance in an attempt to study possible relation of the OS status and egg production. The analysis proved the presence of significant interdependence between the oxidative status of the laying hens and their performance. This was true for all the studied hen’s genotypes. However, the assessed OS markers had not only statistically significant main effects, but also participated in significant interactive effects. In particular, LPO had no statistically significant direct correlation with egg production but participated in a number of significant interactive double and triple effects with the antioxidant enzymes (Table 5). These results strongly indicated that the intensity of the enzymatic antioxidant defence can have a significant effect on egg production. It is considered that the antioxidant enzymes SOD, CAT, and GPx form the first line of defence of the organism against the damaging effect of the ROS [37,39]. This undoubtedly suggests that the induction of pro-oxidative effects by factors associated with the rearing conditions can be specifically compensated by the cell antioxidant system and the degree of this compensation obviously determined the observed significant relations with the egg production of the studied genotypes. ROS trigger the redox-signalling pathways with induction of transcription factors (NF-κB, Nrf2) and specific gene expression with antioxidant response element (ARE)-related synthesis of antioxidant enzymes (SOD, CAT, GPx, glutathione reductase, and glutathione transferase). This regulatory mechanism is considered to be fundamental for the effective antioxidant defence in stress conditions [28]. Therefore, it is not surprising that the activity of antioxidant enzymes is important for the cell welfare and consequently for the normal run of the physiological processes in the organism. Hence, the contribution of both enzyme and non-enzymatic antioxidants for the suppression of OS and sustaining the well-being of birds should not be neglected. This is in line with a number of studies, which have showed that the prevention of OS by supplementation with different antioxidants could improve bird status, performance, and egg quality under stress conditions induced by low- [36] and high-ambient temperature [40,41]. Indeed, the enzyme and non-enzyme antioxidants are proven to work in cooperation to overcome OS and cell injuries, thus facilitating the adaptation to various stresses in avian species [28,37].

## 5. Conclusions

This study reports for the first time results from detailed research of the oxidative status and its effect on the egg-laying ability of six chicken genotypes reared in a free-range system for three periods in two consecutive years. Our data strongly indicated that naturally changing conditions in free-range rearing systems can shift the pro/antioxidant balance of the laying hens, thus raising the risk of OS. This was especially well demonstrated in the response to temperature changes. The chicken genotypes we studied showed differences in their susceptibility/resistance to OS, and this had an effect on their egg production. Generally, our results clearly demonstrated that induced OS is genotypically specific and can play a significant role in determining chicken welfare and egg production in free-range and similar rearing systems. Hence, the high performance of birds in free farming systems would strongly depend on using both highly productive and well adapted to local ambient conditions genotypes.

## Figures and Tables

**Figure 1 animals-12-02650-f001:**
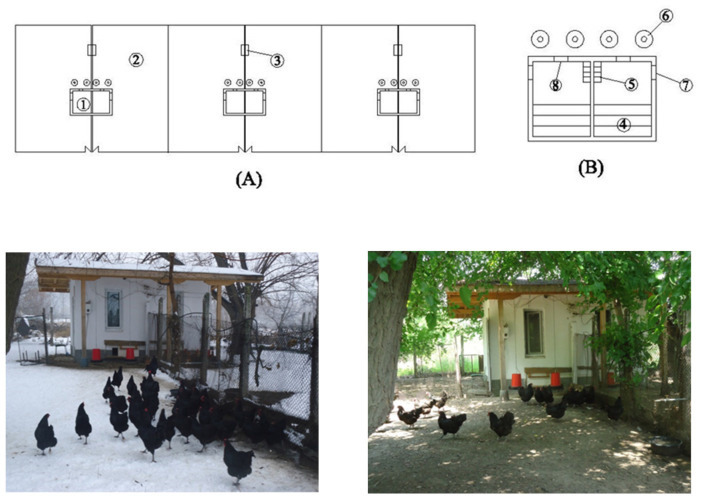
Schematic drawing and photo of the poultry farm: (**A**)—pen ①, fenced yard ②, water drinker ③; pen (**B**)—perches ④, nests ⑤, plastic feeders ⑥, door ⑦, pop-holes ⑧.

**Figure 2 animals-12-02650-f002:**
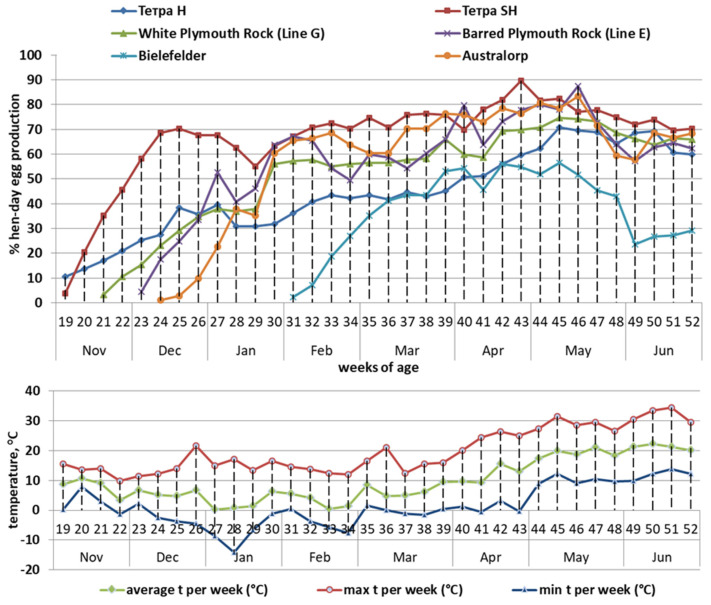
Egg laying performance (%) and ambient temperature (°C) in first experimental year.

**Figure 3 animals-12-02650-f003:**
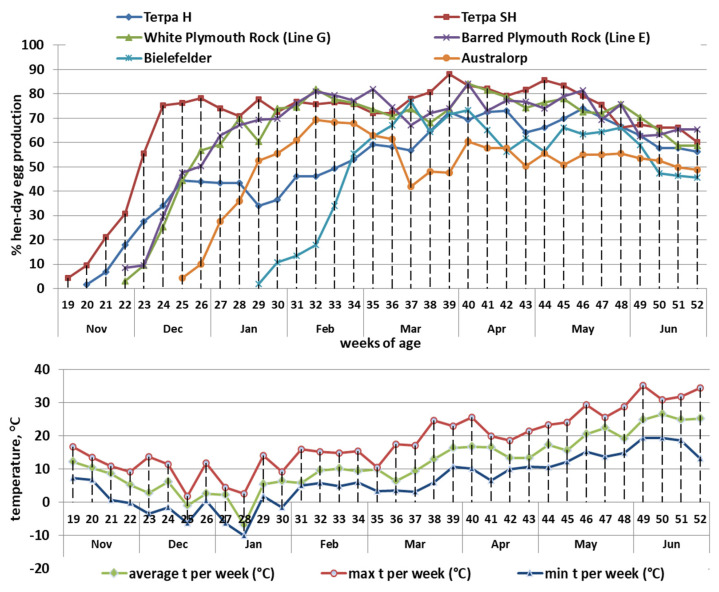
Egg laying performance (%) and ambient temperature (°C) in second experimental year.

**Table 1 animals-12-02650-t001:** Ingredients and nutrient composition of the diets.

Components	Laying Period in Week			
	19–20	21–28	29–40	41–52
Feed ingredients, g/kg				
Corn yellow	336.7	308.7	309.2	320.4
Wheat	336.7	308.7	309.2	320.4
Soybeans meal (440 g crude protein)	120.0	125.0	120.0	100.0
Sunflower expeller (340 g crude protein)	150.0	150.0	150.0	150.0
Sunflower oil	-	12.0	5.0	-
L-lysine	1.1	0.7	0.1	0.5
DL-Methionine	0.7	0.9	0.7	0.7
Sodium chloride	2.6	2.8	2.3	2.0
Limestone	38.0	82.0	96.0	100.0
Dicalcium phosphate	12.0	7.0	5.3	3.8
Premix TB 301 Layers *	2.0	2.0	2.0	2.0
Synergen **	0.2	0.2	0.2	0.2
Calculated composition (per kg)				
Metabolizable energy, MJ/kg	12.1	11.8	11.5	11.5
Crude protein, g	178.0	174.0	171.0	165.0
Crude fiber, g	45.0	44.0	44.0	43.0
Crude fats, g	41.0	51.0	44.0	40.0
Calcium, g	20.0	36.0	38.1	39.2
Phosphorus available, g	6.6	5.6	5.3	3.0
Lysine, g	8.5	8.1	7.5	7.3
Methionine + cysteine, g	7.3	7.3	7.0	6.9

* Premix TB 301 Layers (made in De Heus Koudijs Animal Nutrition Rubensstraat 175, 6717 VE Ede, the Netherlands) in 1 kg contains: vitamin A–5,000,000 IU; vitamin D_3_—1,500,000 IU; vitamin E—4000 mg; vitamin K_3_—500 mg; vitamin B_1_—250 mg; vitamin B_2_—1500 mg; calcium-D-pantothenate—3000 mg; vitamin PP—10,000 mg; vitamin B_6_—500 mg; vitamin B_9_ folic acid—250 mg; vitamin B_12_—10,000 mg; vitamin B_4_—50,000 mg; Fe—20,000 mg; I—400 mg; Cu—2300 mg; Mn—32,500 mg; Zn—25,000 mg; Se—125 mg; antioxidants: propyl gallate—41.7 mg; BHT—41.7 mg; ethoxycuine—41.7 mg; preserving agent: citric acid—0.1 g. ** Synergen—A product of the solid state fermentation of Aspergillus niger (made in Alltech^®^, Lexington, KY, USA).

**Table 2 animals-12-02650-t002:** Weekly ambient temperatures during blood sampling.

Period (Month; Week-Old Hens)	Temperature	I Experiment (°C)	II Experiment (°C)
Cold period (March; 36 weeks of age)	Average	4.82	6.39
	Min	−1.0	−2.8
	Max	21.0	18.0
Thermoneutral period (May; 44 weeks of age)	Average	17.35	17.24
	Min	8.8	8.2
	Max	27.4	26.0
Hot period (July; 52 weeks of age)	Average	21.2	25.6
	Min	13.9	16.0
	Max	34.5	35.8

**Table 3 animals-12-02650-t003:** Oxidative stress markers in different chicken genotypes, years, and periods.

Factors			Oxidative Stress Markers									
Year	Period	Genotype	LPO, Nmoles MDA/mg Hb		tGSH, ng/mg Hb		CAT, U/mg Hb		SOD, U/mg Hb		GPx, U/mg Hb	
I	cold	Tetra H	1.28 ± 0.04	^abc^	738 ± 27	^cdefghi^	0.29 ± 0.03	^ij^	3.29 ± 0.09	^cdef^	38.45 ± 2.57	^abcdefg^
		Tetra Super Harco	1.81 ± 0.08	^ab^	601 ± 110	^fghi^	0.37 ± 0.07	^ghij^	2.61 ± 0.52	^defghi^	41.36 ± 2.91	^abcd^
		White Plymouth Rock	1.56 ± 0.03	^ab^	642 ± 5	^efghi^	0.44 ± 0.09	^ghij^	1.92 ± 0.09	^efghi^	42.00 ± 3.67	^abcd^
		Barred Plymouth Rock	1.21 ± 0.04	^abcde^	528 ± 55	^hi^	0.23 ± 0.02	^j^	2.74 ± 0.22	^defghi^	28.37 ± 2.57	^fghijk^
		Bielefelder	1.24 ± 0.01	^abcd^	585 ± 33	^ghi^	0.41 ± 0.06	^ghij^	3.23 ± 0.54	^cdefg^	43.83 ± 3.85	^abc^
		Australorp	1.05 ± 0.01	^abcde^	490 ± 47	^i^	0.46 ± 0.04	^fghij^	2.73 ± 0.31	^defghi^	50.25 ± 3.11	^a^
		** *m ± SEM* **	** *1.36 ± 0.11* **		** *597 ± 36* **		** *0.37 ± 0.04* **		** *2.75 ± 0.20* **		** *40.71 ± 2.94* **	
	thermoneutral	Tetra H	0.29 ± 0.03	^cde^	897 ± 26	^abcdefghi^	0.30 ± 0.02	^hij^	1.55 ± 0.22	^i^	31.75 ± 3.71	^cdefghij^
		Tetra Super Harco	0.24 ± 0.03	^cde^	793 ± 72	^bcdefghi^	0.45 ± 0.15	^ghij^	1.35 ± 0.03	^i^	16.75 ± 1.09	^kl^
		White Plymouth Rock	0.39 ± 0.01	^cde^	828 ± 23	^bcdefghi^	0.66 ± 0.11	^cdefghij^	1.65 ± 0.18	^hi^	27.50 ± 3.18	^ghijk^
		Barred Plymouth Rock	0.19 ± 0.06	^de^	781 ± 23	^bcdefghi^	0.57 ± 0.03	^defghij^	1.46 ± 0.15	^i^	27.75 ± 2.51	^ghijk^
		Bielefelder	0.25 ± 0.05	^cde^	817 ± 68	^bcdefghi^	0.54 ± 0.04	^defghij^	1.47 ± 0.07	^i^	34.25 ± 3.25	^cdefghi^
		Australorp	0.16 ± 0.01	^e^	804 ± 14	^bcdefghi^	0.57 ± 0.02	^defghij^	1.70 ± 0.47	^ghi^	40.25 ± 3.50	^abcdef^
		** *m ± SEM* **	** *0.26 ± 0.03* **		** *820 ± 17* **		** *0.52 ± 0.05* **		** *1.53 ± 0.05* **		** *29.71 ± 3.23* **	
	hot	Tetra H	1.77 ± 0.14	^ab^	847 ± 75	^abcdefghi^	1.32 ± 0.23	^ab^	5.51 ± 0.55	^a^	48.75 ± 3.02	^ab^
		Tetra Super Harco	1.30 ± 0.24	^abc^	1082 ± 133	^abcde^	1.54 ± 0.13	^a^	4.87 ± 0.24	^ab^	31.75 ± 2.52	^cdefghij^
		White Plymouth Rock	1.91 ± 0.34	^ab^	853 ± 76	^abcdefghi^	0.78 ± 0.07	^cdefgh^	3.17 ± 0.18	^cdefgh^	34.76 ± 1.84	^cdefghi^
		Barred Plymouth Rock	1.91 ± 0.18	^ab^	1198 ± 24	^ab^	0.95 ± 0.10	^bcde^	2.71 ± 0.45	^defghi^	36.50 ± 3.68	^cdefgh^
		Bielefelder	1.01 ± 0.25	^abcde^	773 ± 32	^bcdefghi^	0.58 ± 0.04	^defghij^	4.67 ± 0.39	^abc^	31.75 ± 2.30	^cdefghij^
		Australorp	0.22 ± 0.02	^de^	1105 ± 41	^abcd^	1.01 ± 0.06	^bcd^	3.37 ± 0.14	^bcde^	28.55 ± 1.99	^fghijk^
		** *m ± SEM* **	** *1.35 ± 0.27* **		** *976 ± 71* **		** *1.03 ± 0.14* **		** *4.05 ± 0.46* **		** *35.34 ± 2.91* **	
II	cold	Tetra H	1.16 ± 0.18	^abcde^	597 ± 53	^fghi^	0.53 ± 0.07	^defghij^	3.17 ± 0.14	^cdefgh^	35.30 ± 1.48	^cdefghi^
		Tetra Super Harco	1.98 ± 0.21	^a^	704 ± 43	^defghi^	0.39 ± 0.07	^ghij^	2.54 ± 0.26	^defghi^	32.97 ± 0.78	^cdefghij^
		White Plymouth Rock	1.29 ± 0.18	^abc^	1042 ± 61	^abcdef^	0.40 ± 0.06	^ghij^	2.60 ± 0.19	^defghi^	33.67 ± 1.20	^cdefghi^
		Barred Plymouth Rock	1.55 ± 0.28	^ab^	742 ± 61	^cdefghi^	0.26 ± 0.03	^ij^	2.39 ± 0.22	^defghi^	34.58 ± 1,40	^cdefghi^
		Bielefelder	0.44 ± 0.07	^cde^	658 ± 49	^defghi^	0.43 ± 0.07	^ghij^	2.39 ± 0.22	^defghi^	36.96 ± 0.91	^bcdefgh^
		Australorp	1.02 ± 0.11	^abcde^	888 ± 51	^abcdefghi^	0.74 ± 0.09	^cdefghi^	2.45 ± 0.23	^defghi^	40.96 ± 1.00	^abcde^
		** *m ± SEM* **	** *1.24 ± 0.21* **		** *772 ± 67* **		** *0.46 ± 0.07* **		** *2.59 ± 0.12* **		** *35.74 ± 1.19* **	
	thermoneutral	Tetra H	0.423 ± 0.09	^cde^	911 ± 88	^abcdefghi^	0.69 ± 0.04	^cdefghij^	1.71 ± 0.18	^ghi^	26.38 ± 1.16	^ghijk^
		Tetra Super Harco	0.364 ± 0.06	^cde^	829 ± 30	^bcdefghi^	0.46 ± 0.04	^fghij^	1.75 ± 0.19	^fghi^	13.50 ± 0.45	^l^
		White Plymouth Rock	0.360 ± 0.05	^cde^	1282 ± 113	^a^	0.63 ± 0.03	^cdefghij^	1.84 ± -.13	^efghi^	23.34 ± 1.29	^ijkl^
		Barred Plymouth Rock	0.258 ± 0.05	^cde^	1097 ± 144	^abcd^	0.33 ± 0.02	^ghij^	1.83 ± 0.10	^efghi^	20.63 ± 1.01	^jkl^
		Bielefelder	0.268 ± 0.05	^cde^	1029 ± 86	^abcdefg^	0.52 ± 0.02	^efghij^	2.14 ± 0.15	^efghi^	31.54 ± 1.70	^defghij^
		Australorp	0.320 ± 0.04	^cde^	1158 ± 60	^abc^	0.66 ± 0.02	^cdefghij^	2.19 ± 0.19	^efghi^	35.47 ± 1.27	^cdefghi^
		** *m ± SEM* **	** *0.33 ± 0.03* **		** *1051 ± 67* **		** *0.55 ± 0.06* **		** *1.91 ± 0.08* **		** *25.14 ± 3.2* **	
	hot	Tetra H	1.900 ± 0.34	^ab^	1046 ± 76	^abcdef^	0.97 ± 0.06	^bcde^	3.84 ± 0.14	^bcde^	41.36 ± 1.31	^abcd^
		Tetra Super Harco	1.181 ± 0.11	^abcde^	831 ± 76	^abcdefghi^	1.08 ± 0.04	^abc^	3.26 ± 0.29	^cdef^	25.18 ± 1.31	^hijkl^
		White Plymouth Rock	1.537 ± 0.19	^ab^	706 ± 111	^cdefghi^	0.67 ± 0.04	^cdefghij^	2.75 ± 0.08	^defghi^	29.13 ± 1.20	^efghij^
		Barred Plymouth Rock	1.540 ± 0.36	^ab^	811 ± 71	^bcdefghi^	0.79 ± 0.06	^cdefg^	2.68 ± 0.14	^defghi^	30.35 ± 2.01	^defghij^
		Bielefelder	1.040 ± 0.16	^abcde^	696 ± 56	^defghi^	0.41 ± 0.02	^ghij^	3.11 ± 0.14	^defghi^	33.05 ± 1.02	^cdefghij^
		Australorp	0.864 ± 0.10	^bcde^	980 ± 96	^abcdegh^	0.95 ± 0.07	^bcdef^	2.64 ± 0.24	^defghi^	32.44 ± 1.24	^cdefghij^
		** *m ± SEM* **	** *1.34 ± 0.16* **		** *845 ± 58* **		** *0.81 ± 0.09* **		** *3.05 ± 0.19* **		** *31.92 ± 2.21* **	
**m ± SEM (for I and II years)**			**0.98 ± 0.10**		**844 ± 33**		**0.62 ± 0.05**		**2.65 ± 0.16**		**33.09 ± 1.32**	

Values with different letters are significantly different according to ANOVA and Tukey’s multiple range test. If the letter (s) occupy a more forehead position in the alphabet, then the corresponding indicator has a higher value. The presence of identical letters between two indicators means that there is no significant difference.

**Table 4 animals-12-02650-t004:** Phenotypic correlations (r) between oxidative stress markers.

Indicator	LPO	tGSH	CAT	SOD	GPx
LPO	-	−0.262 **	0.228 **	0.402 **	0.200 **
tGSH		-	0.224 **	−0.02 n.s.	−0.169 *
CAT			-	0.477 **	0.155 *
SOD				-	0.164 *
GPx					-

Significance at *: *p* < 0.05; **: *p* < 0.01; n.s.—No significant difference.

**Table 5 animals-12-02650-t005:** Factorial analysis of variance of the effects of the investigated factors (year, period, genotype) on the oxidative stress markers.

Factors		Source of Variance				
		LPO	tGSH	CAT	SOD	GPx
Year	F	1.369	5.622	3.770	12.401	21.346
	probability	n.s.	*p* < 0.05	*p* < 0.05	*p* < 0.001	*p* < 0.001
Period	F	81.285	22.343	65.846	59.648	9.889
	probability	*p* < 0.001	*p < 0.001*	*p < 0.001*	*p < 0.001*	*p < 0.001*
Genotype	F	5.002	1.754	5.280	3.522	9.744
	probability	*p* < 0.01	n.s.	*p* < 0.001	*p* < 0.01	*p* < 0.001
Year × Period	F	32.408	16.326	30.651	32.605	8.585
	probability	*p* < 0.001	*p* < 0.001	*p* < 0.001	*p* < 0.001	*p* < 0.001
Year × Genotype	F	2.783	2.296	3.400	3.367	8.407
	probability	*p* < 0.05	*p* < 0.05	*p* < 0.001	*p* < 0.001	*p* < 0.001
Period × Genotype	F	20.602	5.128	21.273	13.303	8.792
	probability	*p* < 0.001	*p* < 0.001	*p* < 0.001	*p* < 0.001	*p* < 0.001
Year × Period × Genotype	F	10.812	5.970	14.341	10.193	7.49
	probability	*p* < 0.001	*p* < 0.001	*p* < 0.001	*p* < 0.001	*p* < 0.001

F—Fisher’s statistics; n.s.—No significant difference.

## Data Availability

The datasets used and analysed during the current study is available from the corresponding author on reasonable request.

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
