# Peer review of "Changes in the Oxidative Status of Dual-Purpose Hens Rearing in the Free-Range System during Cold, Thermoneutral and Hot Period"

_animals, 2022, doi:10.3390/ani12192650_

Round 1

Reviewer 1 Report

This is an interesting paper addressing important research questions.

There are few questions to be addressed before the paper can be accepted for publication

1. There were no group replication (repeating an experiment in the second year which would be different in conditions, could not be a replicate. Even if we agree that this is the replication, 2 replicates are not enough to make any judgement.

2. In the study only blood parameters were studied. This is a great limitation of the paper. With poultry, there are usually tissues available for analysis (at least at the end of the study). This would give more information for deeper discussion and understanding.

3. In discussion it would be advisable to give more emphasis to the biological/biochemical/molecular mechanisms determining genotype-releted differences in the antioxidant defences.

4. Line 48- the word "adversely" should be removed, since low stress can have positive effects.

Reviewer 2 Report

The manuscript focus on an interesting aspect of the free range system in poultry. But the manuscript needs some additions and clarifications for an overall improvement. 

Here below you can see an example for a new table that I suggest inserting in M&M instead of lines 127-136 to make it easier to understand:

PERIOD (month)

Parameter

Weekly Temperature

I Experiment-year

(°C)

II Experiment –year

(°C)

COLD (March)

Avg ambient temp

4.82

6.39

Min temp

-1.0

-2.8

Max temp

21.0

18.0

Thermoneutral (May)

Avg ambient temp

17.35

17.24

Min temp

8.8

8.2

Max temp

27.4

26.0

HOT (July)

Avg ambient temp

21.2

25.6

Min temp

13.9

16.0

Max temp

34.5

35.8

All my request are included in the attached pdf file. 

Author Response

Please see the attchment.

Round 2

Reviewer 2 Report

I approve the changes made and only point out to pay attention to the new numbers typed for references within the discussion section, sometimes commas are followed by a blank space (see lines 307 and 342).

Finally, I suggest that the Authors consider the following question: can the Authors indicate a ranking on the best adaptability to the thermal stresses studied, among the six hen genotypes evaluated? If so, provide relevance within the discussion or conclusion.
